# Anti-Neuroinflammatory Potential of a *Nectandra angustifolia* (*Laurel Amarillo*) Ethanolic Extract

**DOI:** 10.3390/antiox12020232

**Published:** 2023-01-19

**Authors:** María Carla Crescitelli, Inmaculada Simon, Leandro Ferrini, Hugo Calvo, Ana M. Torres, Isabel Cabero, Mónica Macías Panedas, Maria B. Rauschemberger, Maria V. Aguirre, Juan Pablo Rodríguez, Marita Hernández, María Luisa Nieto

**Affiliations:** 1Instituto de Biomedicina y Genética Molecular de Valladolid (IBGM), CSIC-Universidad de Valladolid, 47003 Valladolid, Spain; 2Cátedra de Inmunología, Instituto de Ciencias Biológicas y Biomédicas del Sur (INBIOSUR), Universidad Nacional del Sur (UNS), Consejo de Investigaciones Científicas y Técnicas (CONICET), Departamento de Biología, Bioquímica y Farmacia, San Juan 670, 8000 Bahía Blanca, Argentina; 3Laboratorio de Investigaciones Bioquímicas de La Facultad de Medicina (LIBIM), Instituto de Química Básica y Aplicada del NEA, (IQUIBA NEA-UNNE-CONICET), Facultad de Medicina, Universidad Nacional del Nordeste, Corrientes 3400, Argentina; 4Laboratorio de Productos Naturales Prof. Armando Ricciardi (LabProdNat), Instituto de Química Básica y Aplicada del NEA, (IQUIBA NEA-UNNE-CONICET), Facultad de Ciencias Exactas y Naturales y Agrimensura, Universidad Nacional del Nordeste, Corrientes 3400, Argentina

**Keywords:** microglia, neuroinflammation, *Nectandra angustifolia* extract, migration, phagocytosis, ROS

## Abstract

Microglia, the resident macrophage-like population in the CNS, plays an important role in the pathogenesis of many neurodegenerative disorders. *Nectandra genus* is known to produce different metabolites with anti-inflammatory, anti-oxidant and analgesic properties. Although the species *Nectandra angustifolia* is popularly used for the treatment of different types of inflammatory processes, its biological effects on neuroinflammation have not yet been addressed. In this study, we have investigated the role of a *Nectandra angustifolia* ethanolic extract (NaE) in lipopolysaccharide (LPS)-induced neuroinflammation in vitro and in vivo. In LPS-activated BV2 microglial cells, NaE significantly reduced the induced proinflammatory mediators TNF-α, IL-1β, IL-6, COX-2 and iNOS, as well as NO accumulation, while it promoted IL-10 secretion and YM-1 expression. Likewise, reduced CD14 expression levels were detected in microglial cells in the NaE+LPS group. NaE also attenuated LPS-induced ROS and lipid peroxidation build-up in BV2 cells. Mechanistically, NaE prevented NF-κB and MAPKs phosphorylation, as well as NLRP3 upregulation when added before LPS stimulation, although it did not affect the level of some proteins related to antioxidant defense such as Keap-1 and HO-1. Additionally, we observed that NaE modulated some activated microglia functions, decreasing cell migration, without affecting their phagocytic capabilities. In LPS-injected mice, NaE pre-treatment markedly suppressed the up-regulated TNF-α, IL-6 and IL-1β mRNA expression induced by LPS in brain. Our findings indicate that NaE is beneficial in preventing the neuroinflammatory response both in vivo and in vitro. NaE may regulate microglia homeostasis, not only restraining activation of LPS towards the M1 phenotype but promoting an M2 phenotype.

## 1. Introduction

Neuroinflammation is involved in central nervous system (CNS) disorders, such as brain infections, ischemia, trauma and degenerative CNS diseases, and this process is associated with activation of glial cells, astrocytes and microglia, by exogenous and/or endogenous molecules. Microglia is the first cell type to respond to CNS injury and its activation is part of a protective physiological response to remove harmful factors and repair damaged tissue [1,2,3]. However, persistent inflammatory response leads to excessive activation of cells in the CNS and generation of large amounts of regulatory mediators, such as reactive oxygen species (ROS), nitric oxide (NO) and cytokines including IL-6, IL-1β and TNF-α, among others, which further aggravate the progression of the disease [4,5]. Therefore, suppressing immune cell overactivation and its consequent neuroinflammatory response has been recognized as a valuable strategy to improve remission of neurogenic diseases.

The most-used drugs in clinical practice for inflammation are synthetic non-steroidal anti-inflammatory drugs (NSAIDs); however, they have multiple side effects and complications, which weaken the overall potential of this anti-inflammatory treatment [6,7]. Natural plant extracts are sources of high-quality phytochemicals and novel bioactive molecules with important antioxidant and anti-inflammatory effects and could be used in the development of new drugs for the treatment and prevention of inflammatory diseases [8,9,10,11,12]. 

*Nectandra angustifolia* (Schrad.) Nees & Mart., popularly called “yellow laurel”, “river laurel” or “aju’y hû”, is a native plant that can be found largely in Argentina, Brazil, and Uruguay [13]. Some species of the genus *Nectandra* have been used in folk medicine as digestives, purgatives and antispasmodics and for treatment of rheumatism, arthritis and pain [14,15,16]. Moreover, local inhabitants use leaves to ameliorate the local inflammatory effects caused by venomous snake bites [17]. Phytochemical studies have reported the presence of polyphenolic compounds, including flavonoids and lignans, in species of *Nectandra,* which contribute to their anti-inflammatory properties [18]. Recently, some reports have shown anti-inflammatory activities of *Nectandra angustifolia* extracts [19,20,21]. However, after a thorough review of the literature, we have found no studies reporting the therapeutic effect and mechanism of *Nectandra angustifolia* extracts on neuroinflammation.

Therefore, the aim of this study was to evaluate the therapeutic effect of a characterized ethanolic extract of *Nectandra angustifolia* (NaE) on the progression of neuroinflammation through both in vitro and in vivo models and to elucidate its impact on crucial signaling mediators. The present study provides information revealing NaE as an extract with anti-inflammatory actions and suggests a scientific basis for further investigation of NaE against neuroinflammatory conditions.

## 2. Materials and Methods

### 2.1. Reagents

LPS, FITC-dextran and other chemicals were from Sigma Chemical Co. (St. Louis, MO, USA). Hybond-P membrane was from Amersham Biosciences (GE Healthcare Europe GmbH, Barcelona, Spain). DHE was from Molecular Probes (Carlsbad, CA, USA). Cell culture medium and supplements were from Gibco (Gibco BRL, Burlington, ON, Canada).

### 2.2. Plant Material, Preparation of Ethanol Extract and Chemical Characterization

As previously described by Ferrini L et al. [19], aerial parts of *Nectandra angustifolia* were collected in Corrientes, Argentina (27°50′51.9″ S 58°42′46.3″ W). Briefly, the extract was obtained from air-dried leaves by maceration with ethanol at 95° (48 h). After vacuum filtration, the ethanolic extract was evaporated using a rotary evaporator (Büchi R-124). Until further use, the ethanolic extract (NaE) was kept in desiccators under reduced pressure. Chemical composition and HPLC characterization of the extract has already been published, and it includes quercetin, rutin, quercitrin, quercetin-3-β-D-glucoside, quercetin-3-O-neohesperidoside and natsudaidain-3-glucoside [19,20]. NaE was administered 30 min prior to the LPS injection in mice and 1 h before LPS stimulation in BV2 microglia cells.

### 2.3. Cell Culture

The immortalized mouse BV2 microglial cell line, a generous gift from Dr. J.R. Bethea (University of Miami School of Medicine, Miami, FL, USA), was cultured at 37 °C in a humidified atmosphere of 5% CO_2_ in high-glucose Dulbecco’s modified Eagle’s medium (DMEM), supplemented with 100 U/mL penicillin, 100 µg/mL streptomycin, 50 µg/mL gentamicin, 2 mM glutamine, and 10% heat-inactivated fetal bovine serum (FBS) [22]. Cells were serum-starved overnight (o/n) before the experiments. Cells were then left untreated (resting microglia) or stimulated with 0.1 μg/mL of LPS (reactive microglia) at different times in the presence or absence of several doses of the NaE extract specifically indicated in panel A of each figure. 

### 2.4. Viability Assay

The commercial kit Cell Titer 96^®^Aqueous One Solution Cell Proliferation Assay (Promega Corporation, Madison, WI, USA) was used to assess cell viability. Briefly, BV2 microglial cells were seeded in 96-well tissue culture plates and serum-starved for 24 h. Following 30 min of pre-treatment with different doses of NaE (0–50 μg/mL) at 37 °C, cells were incubated with 0.1 μg/mL of LPS, or none, for 24 h. Afterwards, 20 µL of MTT labeling reagent was added to each well, and the plates were incubated at 37 °C for another 4 h. The formazan product formation was measured by recording the absorbance at 490 nm by using a VersaMax^TM^ Tunable microplate reader (Molecular Devices LLC, Sunnyvale, CA, USA). The results were expressed as optical density (OD) values, as an assessment of the number of metabolically active cells. Microglia cell viability was also assessed by trypan blue exclusion.

### 2.5. Flow Cytometric (FC) Analysis

BV2 cells, 5 × 10^6^/flask, were treated with 0.1 µg/mL of LPS for 24 h in the presence or absence of NaE. Cells suspended in cold PBS with 1% BSA were then incubated in the dark with a PE-conjugated CD14 antibody (BioLegend, San Diego, CA, USA) [23]. Following incubation, the cell suspension was centrifuged at 500× *g* for 5 min, washed twice, and resuspended in PBS-BSA 1%. Cells were then analyzed on a flow cytometer (Gallios^TM^, Beckman Coulter, Miami, FL, USA), using the Beckman Coulter Kaluza Analysis Software. The mean fluorescence intensity (MFI) of CD14-positive cells was measured. 

### 2.6. Western Blot (WB) Analysis

After treatment, cells were washed twice with PBS and homogenized in lysis buffer (20 mM Tris-HCl (pH 7.4), containing 150 mM NaCl, 0.5% Triton X-100, 1 mM Na3VO4, 150 mM NaF, 1 mM phenylmethylsulfonyl fluoride and a protease inhibitor mixture (Sigma-Aldrich, Madrid, Spain) at 4 °C, and diluted 2:1 in 3x Laemmli’s sample buffer. The amount of protein was assayed using the Bradford reagent, and equal amounts of protein extracts (30 μg) were separated by SDS-PAGE and transferred to polyvinylidene difluoride membranes. They were then briefly stained with Ponceau S to confirm equal loading and transfer of protein. Membranes were blocked for 4 h using 5% nonfat dried milk or 3% BSA (for phospho-protein analysis) in TTBS (20 mM Tris, 150 mM NaCl, and 0.1% Tween 20) at room temperature. After that, they were incubated for 18 h at 4 °C with primary antibodies against COX-2 (Cayman Chemical, Ann Arbor, MI, USA); iNOS (BD Biosciences, Palo Alto, CA, USA); HO-1 (StressMarq Biosciences Inc., Victoria, British Columbia, Canada); YM-1 (StemCell Technologies, Saint Égrève, France); Keap-1, phospho-NFkB-p65 (p-NFkB-p65, Ser536), phospho-p44/42 MAPK (p-ERK1/2, Thr202/Tyr204), phospho-p38 MAPK (p-p38, Thr180/Tyr182), phospho-JNK1/2 (Thr183/Tyr185) and phospho-P70S6 kinase (p-P70S6K, Thr389) (Cell Signaling Technology, Inc, Danvers, MA, USA); NLRP3 (Novus Littleton, CO, USA); and actin (Sigma Chemical Co, St Louis, MO, USA). After washing, membranes were incubated with the corresponding secondary antibody (1:2000, *v*/*v*; Citiva, Brooklyn, NY, USA) conjugated with horseradish peroxidase (HRP) at room temperature for 1 h. Signals were detected using the enhanced chemiluminescence (ECL) system (Millipore, Burlington, MA, USA). Briefly, blots were incubated with the ECL reagents for 1 min, and HRP catalyzes the oxidation of luminol. Then, chemiluminescence was detected photo-graphically by exposure on an X-ray film (Hyperfilm, Amersham International, UK). The band intensity was quantified using ImageJ Software, and β-actin was used as an internal control. Western blot bar graphs can be found as Appendix A.

### 2.7. Measurement of Intracellular Superoxide Radicals

Dihydroethidium (DHE) staining was used to determine superoxide anion production. DHE reacts with O_2_^•−^ to produce the fluorescent product oxyethidium, which binds to DNA and causes an increased fluorescent intensity in cell nuclei. Intracellular superoxide was measured as described previously [24]. 5 × 10^6^ cells were treated with 50 μg/mL of the extract, in the presence or absence of 0.1 μg/mL of LPS for 24 h. After that, they were incubated with 2 µM of DHE for 30 min at 37 °C, scraped, washed and resuspended in ice-cold PBS. The fluorescent signals were analyzed by recording fluorescence in a Gallios^TM^ flow cytometer (Beckman Coulter, Miami, FL, USA).

### 2.8. Measurement of Lipid Peroxidation

Lipid peroxidation was assessed through evaluation of the end-products of lipid peroxidation of 4-Hydroxynonenal (4-HNE) and Malondialdehyde (MDA) [25]. MDA + 4-HNE levels were determined using a commercial kit (BQCkit-Bioquochem, Gijón, Spain) according to the manufacturer’s instructions provided with the reagent kits. Briefly, BV2 cells were treated with different doses of the extract, in the presence or absence of 0.1 μg/mL of LPS. After 24 h, cells were homogenized at 4 °C in 20 mM Tris buffer (pH 7.4) and centrifuged for 10 min at 2000× *g* and 4 °C to remove intact cells. The assay was performed in the cell lysate. In this assay, an indol (Reagent A) reacts quickly with MDA and HNE in acidic medium, yielding a chromophore (C) with a high molar extinction coefficient at its maximal absorption wavelength of 586 nm. The absorbance was determined on a microplate reader (VersaMaxTM, Molecular Devices LLC, Sunnyvale, CA, USA).

### 2.9. NO Assay

Levels of nitrite, a stable breakdown product of NO, were measured with a Griess Reagent System (Promega, Madison, WI, USA) in the culture medium (supernatant). The BV2 cells were seeded in a 24-well plate (n = 3) and, after 24 h of LPS treatment in the presence or absence of NaE, the supernatant was collected. Briefly, 150 μL culture medium was mixed with an equal volume of the Griess reagent (0.1% naphthylethylenediamine hydrochloride and 1% sulfanilamide in 5% phosphoric acid) in a 96-well plate and incubated for 30 min in the dark at room temperature. The absorbance was determined on a microplate reader (VersaMax^TM^, Molecular Devices LLC, Sunnyvale, CA, USA) at 548 nm.

### 2.10. Enzyme-Linked Immunosorbent Assay (ELISA)

Secreted TNF-α, IL-1β, IL-10 and IL-6 protein levels were measured in a cell-cultured medium of BV2 cells incubated with 0.1 µg/mL of LPS for 24 h at 37 °C, in the presence or absence of the NaE extract. Commercial ELISA kits (Invitrogen Waltham, MA, USA) were used according to the manufacturer’s instructions.

### 2.11. Scratch Test Assay

Cells were seeded onto 24-well plates at a density of 1 × 10^5^ cells/well. At 100% confluency, a scratch was made perpendicular to the well using a pipette tip against a ruler [26,27]. After washing with PBS to remove debris from the damaged cells, serum-free DMEM was added. Then, cells were treated with no LPS (control) or with 0.1 μg/mL of LPS in the presence or absence of different doses of NaE. The cells were then fixed and stained with Giemsa dye for 30 min at room temperature. Images were obtained with a phase-contrast microscope (Nikon Eclipse TS100 microscope, Tokyo, Japan) using 40× magnification at 0 h, 3 h and 24 h post-scratch to observe scratch healing. The number of cells migrating into the gap was counted using Image J software (U.S. National Institutes of Health, Bethesda, MD, USA) [28]. 

The migrating cells in control conditions were normalized to 1.00. Results are presented as “relative migration of cells”, which was calculated as: migration number of experimental group/migration number of control group.

### 2.12. Phagocytosis Assay

BV2 microglia cells seeded in 25 cm^2^ flasks were incubated with 0.1 µg/mL of LPS at 37 °C, in the presence or absence of the NaE extract. After 24 h, the phagocytic ability of the cells was measured using yellow-green (YG)-Fluoresbrite^®^-carboxylate-modified microspheres (1 µm) (Polysciences, Inc., Hirschberg an der Bergstrasse, Germany). Briefly, cells were exposed to 108 particles/mL of YG-carboxylate-modified microspheres for 2 h at 37 °C. Non-internalized particles were removed by washing vigorously three times with cold PBS (pH 7.4). Cells were then analyzed by flow cytometry for uptake of fluorescent beads. The mean fluorescence intensity (MFI) of YG-carboxylate-microsphere-positive cells was recorded with a Gallios^TM^ (Beckman Coulter, Miami, FL, USA). Cultures without YG-carboxylate-modified microspheres were used as background (blank wells). 

### 2.13. Animals

Male BALB/c (25–30 g) mice were provided by the animal house of Facultad de Medicina, Argentina—UNNE. All animals were housed under standard laboratory conditions (room temperature: 25.0 ± 2.0 °C, relative humidity: 55–65% and 12 h light/dark cycle) with free access to food and water. Animal treatments were in strict accordance with international ethical guidelines concerning the care and use of laboratory animals. We received approval from the Animal Care Committee of the Facultad de Medicina Universidad Nacional del Nordeste (Protocol #007-2021 CICUAL Fac. Med. -UNNE).

Mice were divided into 5 groups (n = 6 per group): control (vehicle injection), LPS (3 mg/kg i.p.), NaE (5 mg/kg i.p.) + LPS (3 mg/kg i.p.), NaE (50 mg/kg i.p.) + LPS (3 mg/kg i.p.) and dexamethasone (2 mg/kg i.p.) + LPS (3 mg/kg i.p.). Dexamethasone was used as a suitable anti-inflammatory control drug as previously described [29]. The LPS group and dexamethasone + LPS group also received the same vehicle injection (1% DMSO, 10% ethanol, 10% PEG 400 and saline). *Nectandra angustifolia* extract (NaE) was administered 30 min prior to the LPS stimulus, as in previous studies [19]. After 3 h of LPS injection, mice were sacrificed, and brain was collected and conserved for subsequent analysis.

### 2.14. Real-Time Polymerase Chain Reaction Analysis

Total RNA from brain tissue was extracted using TRI reagent (Sigma, Darmstadt, Germany) according to the manufacturer’s instructions. Genomic DNA was degraded using Turbo DNAse (Thermo Scientific, Waltham, MA, USA). The reverse transcription of RNA was performed using M-MLV transcriptase (Sigma, Darmstadt, Germany) according to the manufacturer’s instructions. Gene expression was quantified in the ECO^®^ Real-Time PCR System (Illumina, San Diego, CA, USA) using HOT FIREPol^®^ EvaGreen^®^ qPCR mix (Solis Byodine, Tartu, Estonia) according to the product’s data sheet. Primers used for IL-1β forward 5′-GAAATGCCACCTTTTGACAGTG-3′ reverse 5′- TGGATGCTCTCATCAGGACAG-3′; IL-6 forward 5′- CTGCAAGAGACTTCCATCCAG-3′ reverse 5′- AGTGGTATAGACAGGTCTGTTGG-3′; TNF-α forward 5′- CAGGCGGTGCCTATGTCTC-3′ reverse 5′-CGATCACCCCGAAGTTCAGTAG-3′ and for GAPDH forward 5’-AGGTCGGTGTGAACGGATTTG-3′ reverse 5′-TGTAGACCATGTAGTTGAGGTCA-3′ (Thermo Fisher Scientific, Waltham, MA, USA) [19,30]. The relative expression of IL-1β, IL-6 and TNF-α were normalized to that of GAPDH and calculated using the ΔΔCt method [31].

### 2.15. Statistical Analysis

All data were expressed as the mean ± SD and analyzed by one-way analysis of variance (ANOVA) followed by post-hoc comparisons (Bonferroni test) using the GraphPad Prism Version 5 software (San Diego, CA, USA). *p* < 0.05 was considered statistically significant.

## 3. Results

### 3.1. Effects of NaE on the Inflammatory Response Induced by LPS on BV2 Cells

To determine an appropriate concentration for the NaE extract, we first investigated its cytotoxic effect by measuring the cell viability of BV2 microglia cells using the MTT assay, as detailed in Figure 1A. We assayed a series of concentrations ranging from 10 μg/mL up to 50 μg/mL based on data in the literature [19]. As shown in Figure 1B, none of the concentrations affected BV2 cell viability, since more than 95% of cells treated with NaE for 24 h survived (*p* > 0.05; F = 0.823). Furthermore, as we used LPS to induce microglial inflammatory responses, the effect of LPS treatment on cell viability was also evaluated. Our experimental data demonstrate that the presence of LPS at the concentration of 0.1 μg/mL neither affected BV2 cells’ viability nor did it interfere with the effect of NaE on cell survival (*p* > 0.05; F = 0.963). Representative micrographs of NaE-treated cells are shown in Figure 1C.

Next, to evaluate the role of the NaE in regulating pro-inflammatory signaling pathways activated in neuroinflammatory processes, we examined the effect of NaE in LPS-induced reactive BV2 microglia following the scheme in Figure 2A. Using Western blotting we assessed the phosphorylation profile of intracellular key signaling kinases. Figure 2B shows that the phosphorylation of Erk1/2, JNK1/2, and p38 MAPKs as well as P70S6K was promoted by the inflammogen LPS and inhibited by the NaE in a dose-dependent manner.

Since NF-κB and the inflammasome NLRP3 are the main controllers of the transcription of pro-inflammatory mediators and cytokines, we further investigated whether NaE inhibits its activation in LPS-treated cells. As shown in Figure 2C, when BV2 microglial cells were stimulated with LPS, the expression of NLRP3 as well as NF-κB-p65 phosphorylation was increased compared with the control group, and pre-treatment with NaE markedly reduced the response induced by LPS.

Next, we examined the effects of NaE on the induction of proteins such as COX-2 and iNOS, as well as the levels of NO, a lipophilic free radical released into the cell-cultured supernatant, under inflammatory conditions (Figure 3). Compared with the control group, the expression of the inflammatory proteins COX-2 and iNOS increased in the LPS-treated BV2 group but decreased when cells were pre-treated with NaE in a dose-dependent manner (Figure 3B). In resting conditions (no LPS), pre-treatment of cells with the NaE extract did not impact the expression of these pro-inflammatory factors. Similarly, the production of the signaling molecule NO was increased in the LPS group compared with the control group, and these levels were reduced in the presence of NaE in a concentration-dependent manner. In consonance, NaE alone did not induce NO production (Figure 3C). 

Based on evidence supporting that NO, MAPK family kinases and redox-sensitive transcription factors such NF-κB are involved in the modulation of the phase II antioxidant heme oxygenase-1 (HO-1), we investigated the effects of NaE on the expression of this inducible enzyme with immunomodulatory functions [32,33]. The results indicated that LPS enhanced HO-1 protein levels, and the combination of different doses of the extract with LPS reached almost the same effect as LPS alone (Figure 3D). Moreover, the induction of HO-1 was not affected by pre-treatment with NaE alone. In addition, protein expression levels of the antioxidant SOD2 were not affected by any treatment. 

We also examined, both in resting and reactive microglial cells, the expression levels of Keap-1, a crucial inhibitor protein of the transcription factor Nrf2, which also upregulates HO-1 in different cells and tissues [34]. As shown in Figure 3E, the exposure of control cells to the extract did not affect the constitutive levels of the Keap-1 protein, and the marked Keap-1 reduction observed in LPS-treated cells was not amended by the presence of NaE. Hence, this evidence might indicate that this signaling mediator is not involved in the anti-inflammatory and anti-oxidant effect of NaE. 

Next, commercial ELISA kits were used to detect the effect of NaE on LPS-induced inflammatory cytokines production in BV2 cells (Figure 4). Compared with the untreated control group, concentration levels of the cytokines IL-6, IL-1β and TNF-α were significantly increased in the LPS -stimulated group, while pre-treatment with NaE dramatically reduced the expression of these pro-inflammatory factors. In contrast, IL-10 concentration was markedly lower in the LPS-stimulated BV2 group than in the control group, and NaE pre-treatment significantly increased its expression levels (Figure 4B and Appendix A).

The M2 polarization state is characterized by a decreased production of molecules of the pro-inflammatory response, while secretion of anti-inflammatory cytokines such as IL-10 is favored. Therefore, we also examined the effect of NaE on CD14 and YM-1 expression, as markers of M1 and M2 states, respectively (Figure 4C,D). Although there was some CD14 expression in resting BV2 cells, upon LPS stimulation CD14 increased and NaE pre-treatment attenuated this enhancement.

On the other hand, pre-treatment of BV2 cells with the NaE caused a marked up-regulation of YM-1 protein expression in a dose-dependent manner, both in the absence or presence of LPS.

Finally, since microglia is able to change its morphology in response to extracellular signals [35], we examined under a phase contrast microscope whether NaE was able to modulate the morphological changes of LPS-treated BV2 cells. As shown in Figure 4E, resting BV2 cells were mainly spindle-shaped, with small cell bodies and long processes, and, once activated with LPS, cells displayed roundish or amoeboid shapes with fewer and shorter branches. The presence of NaE did not affect the cell morphology of resting cells compared with the untreated controls, while, in LPS-treated cells, NaE pre-treatment attenuated LPS-induced cell swelling and the ameboid phenotype, and cells showed increased ramifications (Figure 4E).

### 3.2. NaE Modulates LPS-Mediated BV2 Cells’ Oxidative Stress

We further investigated whether NaE reduced ROS generation and oxidative damage in LPS-stimulated microglia cells (Figure 5A). The data demonstrated that LPS treatment significantly increased the levels of superoxide anion (O_2_^−^) production (Figure 5B,C), as well as lipid oxidative damage (MDA + HNE levels) (Figure 5D) in BV2 cells. NaE pre-treatment significantly reduced the levels of both superoxide anion and lipid peroxidation.

### 3.3. NaE Modulates LPS-Mediated BV2 Cell Migration

The migratory function of microglia is a characteristic of the inflammatory responses during the early phases of neurodegeneration [36]. We examined whether NaE could regulate cell migration induced by LPS administration using the scratch wound-healing assay, as detailed in Figure 6A. At 0 h post-scratch, cells were similarly distributed in all groups, with minimal cells present in the scratched area (Appendix A). As shown in Figure 6B, LPS stimulation promoted the migration of BV2 cells in the scratched area compared with the control group, both at 3 h and 24 h, and pre-treatment with the NaE extract significantly inhibited LPS-enhanced cell migration (*p* < 0.001). Representative images showing that cells pre-treated with NaE exhibited greater difficulty in migrating, compared with those stimulated with LPS (Figure 6C and Appendix A). Moreover, application of NaE alone to resting cells, also reduced, in a dose-dependent manner, the migratory potential of microglial cells, as the free area of the scratch remained even wider than the resting conditions without treatment. Hence, these findings suggested that NaE blocked microglial migration.

### 3.4. Effect of NaE on BV2 Cell Phagocytosis

Another key function of microglia under inflammatory conditions, but also in noninflammatory environments, is debris clearance, including phagocytosis of apoptotic neurons (efferocytosis) [37]. In this study, to analyze the effect of the NaE extract on BV2 cell phagocytosis, YG-latex beads were used followed by flow cytometer analysis. We also examined whether the switch from resting to reactive microglia caused any changes in the actions of the NaE extract. Firstly, we observed that incubation with LPS as a stimulator of inflammatory processes enhanced the phagocytic capacity in BV2 cells, compared with the control resting cells, as shown in the histograms and represented in the corresponding graph (Figure 6D,E). It can be inferred from the figures that pre-treatment with the NaE extract did not affect the phagocytic activity, neither in reactive nor in resting BV2 cells.

### 3.5. NaE Modulates In Vivo Neuroinflammation

Next, we aimed to explore whether NaE could ameliorate neuroinflammation in an animal model. Given that neuroinflammation is mediated by the secretion of proinflammatory cytokines such as TNF-α, IL-1β and IL-6 derived from the activation of microglia [38], we used LPS-injected mice, a well-known neuroinflammation model showing induction of pro-inflammatory cytokine in the brain [30,31,32,33,34,35,36,37,38,39,40,41]. Balb/c mice were i.p.-injected once with LPS and, after 3 h, IL-1β, IL-6 and TNF-α expression was measured in brain tissue homogenates by RT- qPCR (Figure 7). As expected, these proinflammatory cytokines showed higher levels in the brain of LPS-injected mice compared with those of control mice. In line with the in vitro findings, the levels of these three cytokines were significantly lower in the brain of mice treated with NaE before LPS injection. NaE ameliorated the neuroinflammatory response in a dose-dependent manner. The 50 mg/kg dose of NaE induced an effect comparable to pre-treatment with dexamethasone for the three cytokines studied.

## 4. Discussion

In this study, we investigated the anti-neuroinflammatory effect of an ethanolic extract from *Nectandra angustifolia* (NaE). Our findings showed that NaE could inhibit microglia activation in BV2 microglial cells exposed to LPS and reduce cytokine levels in the brain of LPS-injected mice.

Plants of the genus *Nectandra* have long been used in traditional medicine as antifungal, antimalarial and analgesic treatments, and it also exhibits spasmolytic and antineuralgic properties, among others. The species *Nectandra angustifolia* has been reported to show activity for the treatment of rheumatism, arthritis and pain, and as an antivenom [13,14,15,16]. The presence of different secondary metabolites, including flavonoids and phenolic acids, contributes to the biological properties already described for NaE [18,19]. Although phytochemicals have been shown to have potential protective effects in various neurological disorders, the anti-neuroinflammatory properties of this bioactive extract have not yet been investigated [42]. 

Neuroinflammation plays major roles in the pathogenesis of numerous neurological disorders, contributing to deleterious effects on the CNS. The neuroinflammatory response is characterized by increased activity of microglial cells, which are the first responders to CNS insults and play a pivotal role in both physiological and pathological conditions [1,2,3]. Microglial over-activation leads to an excessive secretion of inflammatory factors that often exert adverse effects [43]. In this study, we found that NaE effectively exerted an inhibitory effect against neuroinflammation in LPS-induced reactive BV2 microglia.

Following LPS treatment, we observed that activated microglia released several cytotoxic substances, such as ROS (superoxide, NO) and proinflammatory cytokines, including TNF-α, IL-1β and IL-6. Moreover, the expression of M1 markers, such as iNOS, COX-2 and CD14, increased. In contrast, the expression of the anti-inflammatory cytokine IL-10 decreased, and the expression of YM-1 remained unchanged, both being markers of the M2 phenotype. In this study, pre-treatment with different concentrations of NaE inhibited the activation and M1 polarization of BV2 microglia cells, while promoting transition into M2 polarization: NaE reduced the expression levels of iNOS and COX-2 in LPS-stimulated cells, as well as the presence of oxidized lipids and ROS and the secretion of IL-1β, IL-6, and TNF-α, while the expression of YM-1 and production of IL-10 increased. Moreover, our findings from the in vivo study in the model of neuroinflammation, elicited by injection of LPS into mice, also demonstrated that NaE was effective in lowering pro-inflammatory cytokine expression in brain tissue. This model is characterized by increased brain levels of TNF-α, IL-6 and IL-1β in LPS-challenged mice, and NaE administration prevented this abnormal cytokine expression [41]. These in vivo actions are in consonance with those previously reported in other inflammation models [19]. However, additional in vivo experiments are needed in order to validate our preliminary findings. This preclinical investigation deserves not only a study of the prophylactic/preventive actions of NaE, but also an examination of its potential curative effects by administration of the extract before/after disease onset in models of chronic neuroinflammation.

It is of note that NaE exhibited anti-inflammatory properties similar to traditional drugs such as dexamethasone [44,45]. These effects, showing the NaE-mediated inhibition of brain pro-inflammatory cytokines build-up and, thus, its ability to inhibit neuroinflammation, might suggest that extract component(s) can penetrate the blood–brain barrier, supporting the possible use of NaE as a protective agent in the CNS. However, although constituents of the extract, such as flavonoids and polyphenols, can penetrate the blood–brain barrier, at present, we are not able to postulate a precise mechanism of NaE for this behavior, which is also beyond the purposes of this study [46,47]. Further understanding of the modulation of NaE in the animal model will unveil NaE targets, as well as the role of microglia and other CNS cells in the process.

The signaling pathways reported to be involved in the activation of microglia towards the classic activated phenotype M1 comprise MAPKs (ERK/P38/JNK), NF-κB and NLRP3 [48]. These signaling mediators play an important role in the regulation of inflammation and, therefore, in the induction of proinflammatory factor expression. In the present study, we observed that NaE pre-treatment inhibited LPS-induced activation of these signaling kinases and transcription factors, thus suggesting that the anti-neuroinflammatory effect of NaE may also be attributed to the inhibition of these pathways. Additionally, LPS-induced neuroinflammation in microglial cells is associated with the inhibition of autophagic flux through the activation of the PI3KI/AKT/mTOR pathway: enhanced microglial autophagy downregulates LPS-induced neuroinflammation [49]. In line with these findings, we observed that P70S6K phosphorylation, a mediator in the PI3K/mTOR signaling pathway, was also abolished in cells exposed to NaE before LPS stimulation. However, the role of NaE in modulating the autophagy flux needs supplementary exploration. Interestingly, positive effects of a variety of plant extracts and dietary polyphenols due to their ability to induce autophagy stimulation mediated by mTOR activation have been reported [50].

Moreover, in addition to the modulation of the NF-κB/MAPK and NLRP3 signaling cascades, the upregulation of the Nrf2 pathway also exhibits antioxidant and anti-inflammatory effects through the regulation of genes encoding cyto-protective enzymes. Recently, it has been reported that a wide range of bioactive compounds derived from natural sources, apart from inhibiting NLRP3/NF-κB signaling, modulate the Keap1/Nrf2/HO-1 pathway [51,52]. LPS signaling is known to induce the degradation of Keap-1, which regulates the activity of the transcription factor Nrf2 [53]. In this study, we also made a first approach towards assessing potential changes in the endogenous defense system due to treatment with NaE, focusing on the HO-1 and SOD proteins, and we observed that the presence of NaE did not affect their expression levels under basal conditions (in resting microglia), neither did it modulate the response induced by the inflammatory stimulus, LPS. Additionally, we also found that NaE was not able to stimulate Keap-1 degradation by itself. It has been shown that phytochemical activators of Nrf2 have the capability of modifying highly reactive cysteine residues of Keap1, which then loses its ability to target Nrf2 for degradation, resulting in its stabilization, nuclear translocation, and transcriptional induction of Nrf2-target genes [54]. Therefore, it may be hypothesized that the NaE extract lacks this competence.

On this basis, we might support the existence of a Keap-1/HO-1 independent pathway. However, the Keap1/Nrf2 pathway is a complex signaling pathway, and many factors regulating the stability or activity of Nrf2 should be considered [55,56]. Thus, modulation of this cascade not only includes Nrf2 translocation to the nucleus, but also changes in the levels of cytoplasmic proteins interfering with the Keap1–Nrf2 interaction (e.g., p21 or p62), or the availability of Nrf2 binding partners at the nucleus; for a review, see Jaramillo et al. [57]. These factors should be borne in mind to elucidate the involvement of the Nrf2 pathway in our experimental conditions. Moreover, other antioxidant response elements, such as NQO1, glutathione S transferases (GST), antioxidants and related proteins (thioredoxins, γ-GCS), glutathione peroxidase and glutathione reductase should be also taken into consideration. Gaining deep insights into the actions of the extract on this stress sensing pathway remains a challenge for the future.

In addition, NaE, as a phenolic extract, has been previously demonstrated by Ferrini et al. [19] to possess free radical scavenger activity, which might explain the antioxidant effect observed in BV2 cells stimulated in its presence. 

Key features of microglial activation include phagocytosis and cell migration. Microglial migration is a key step in the inflammatory response. The M1 phenotype mainly migrates towards the site of neuroinflammation, where it secretes inflammatory factors, which may act as chemoattractants, amplifying inflammation and promoting neurodegeneration [36]. Therefore, inhibition of cell migration is an important strategy to suppress inflammatory responses. Consistent with previous reports, we observed, with the aid of wound-healing assays, that LPS treatment increased BV2 cell migration and NaE pre-treatment markedly suppressed BV2 cell mobility [58]. We also found that LPS transformed microglial morphology from a polarized baseline state to an ameboid-like state, and NaE pre-treatment prevented these changes. It is well accepted that LPS-induced morphological and functional changes are related to cytoskeletal rearrangement, F-actin being the major contributor, and research on natural products has documented their efficacy in ameliorating these cytoskeletal organization variations [59,60,61]. Since NaE affected the migration capacity of BV2 cells, to investigate NaE actions regarding actin cytoskeletal morphology represents a forthcoming challenge. 

Additionally, we observed that the phagocytic function was not affected by the NaE extract. Given that neurodegenerative processes may generate debris (e.g., dead neurons) and that debris accumulation exacerbates neuroinflammation by over-activating microglia, the fact that in the presence of NaE the phagocytic capabilities remained unchanged might indicate that this essential component of the brain’s regenerative response against neuroinflammation is preserved [62]. 

## 5. Conclusions

In summary, our observations demonstrated the anti-inflammatory and anti-oxidant properties of NaE in LPS-induced neuroinflammation. Our data support the hypothesis that NaE exerts its protective effects by promoting microglial polarization into the M2 phenotype, whilst suppressing the M1 phenotype and restraining cell mobility, without affecting its phagocytic capabilities. In addition, the study reveals in vivo activities of NaE by reducing cytokines in the brains of mice that were challenged by LPS. Based on these findings, potential protective effects of NaE in the pre-treatment of neuro-inflammatory diseases are plausible and make its components an attractive research option for the development of drugs based on it in the near future.

## Figures and Tables

**Figure 1 antioxidants-12-00232-f001:**
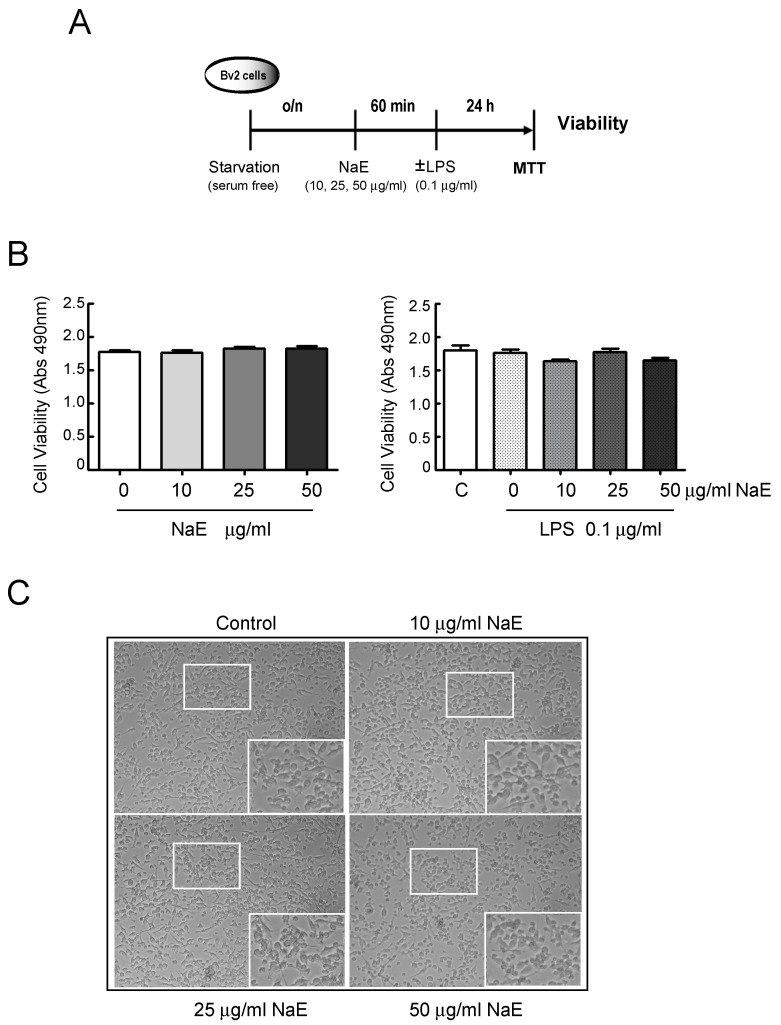
Effect of NaE extract on the viability of BV2 microglia. (**A**) Experimental steps. (**B**) BV2 cells were treated with the indicated doses of the NaE extract in the absence or presence of 0.1 μg/mL LPS for 24 h. Cell viability was measured using an MTT assay. Values are expressed as the mean ± SD of the three independent experiments. (**C**) Representative photomicrographs of cells incubated with medium (control) or NaE (10, 25 and 50 µg/mL) for 24 h. Images were taken at 100× and 40× magnification. N = 3.

**Figure 2 antioxidants-12-00232-f002:**
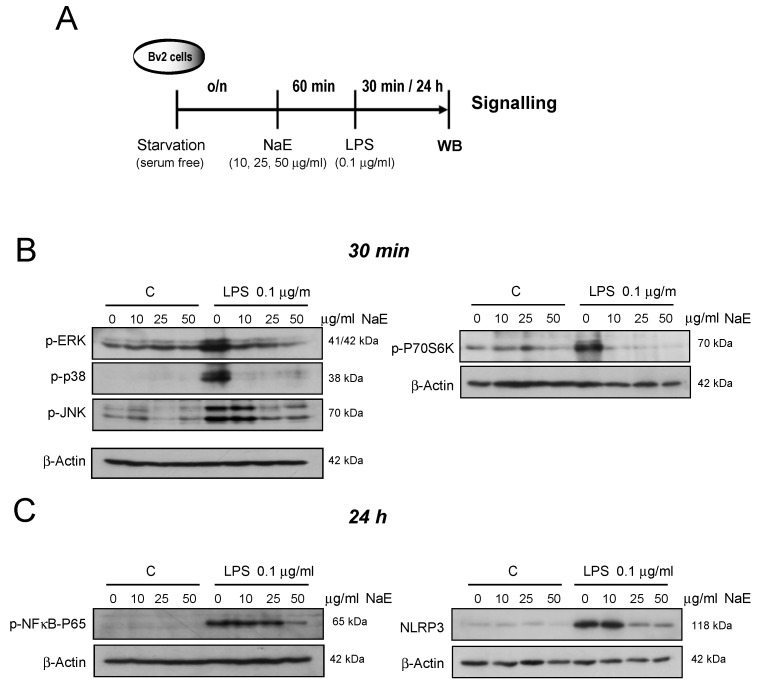
Effect of NaE extract on LPS-modulated signaling proteins in BV2 microglia. (**A**) Experimental steps. Cells were pre-treated with the indicated doses of the NaE extract for 1 h at 37 °C, and then were stimulated with or without 0.1 μg/mL of LPS for (**B**) 30 min or (**C**) 24 h. (**B**) Activation of the MAPKs family was verified using specific antibodies of phosphorylated ERK, p38, JNK and P70S6K proteins. (**C**) Expression levels of NLRP3 and phosphorylated NFkB-p65 were detected using specific antibodies. β-actin was used as the internal control. N = 4. Western blot bar graphs are in Appendix A.

**Figure 3 antioxidants-12-00232-f003:**
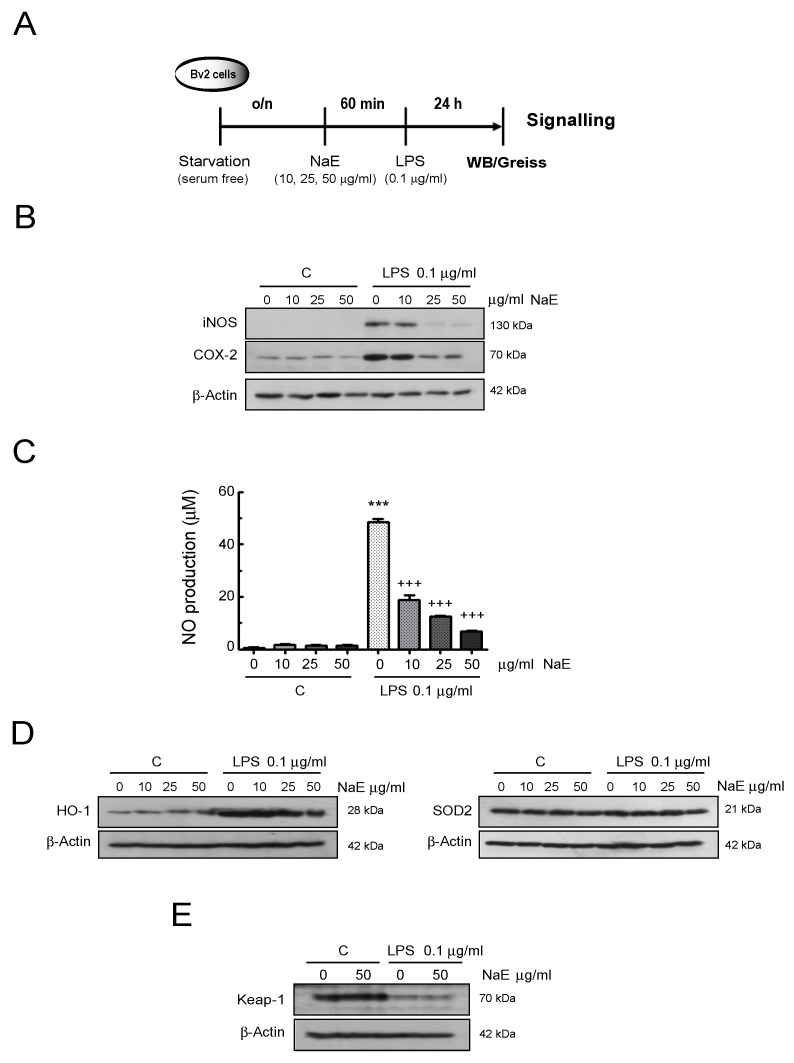
Anti-inflammatory effect of NaE on BV2 cells. (**A**) Experimental steps. BV2 cells were incubated for 1 h at 37 °C with the indicated doses of the NaE extract and then stimulated with or without 0.1 μg/mL of LPS for 24 h. (**B**) Expression of COX-2 and iNOS analyzed by western blot using specific antibodies. (**C**) NO accumulation in the cell culture supernatants analyzed by a Griess assay. Values are means ± SD (n = 3–5). *** *p* < 0.001 vs. controls; and +++ *p* < 0.001 vs. LPS-treated cells. Expression of (**D**) HO-1 and SOD2 as well as (**E**) Keap-1 analyzed by western blot using a specific antibody. Western blot bar graphs are in Appendix A.

**Figure 4 antioxidants-12-00232-f004:**
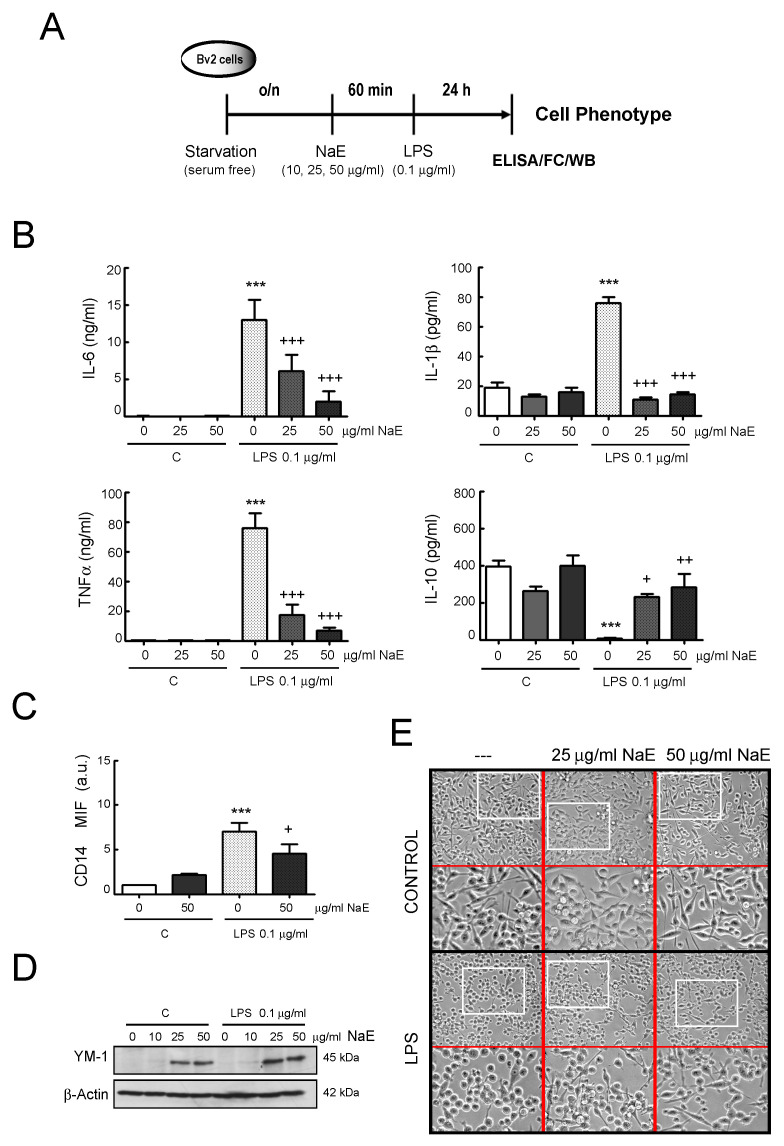
NaE extract modulates the M1 phenotype induced by LPS in BV2 cells. (**A**) Experimental steps. BV2 cells were incubated for 1 h at 37 °C with the indicated doses of the NaE extract, and then stimulated with or without 0.1 μg/mL of LPS for 24 h. (**B**) Levels of IL-6, TNF-α, IL-1β and IL-10 were determined in the culture supernatants using commercial ELISA kits. Values are means ± SD (n = 3–5). *** *p* < 0.001 vs. controls; and +++ *p* < 0.001, ++ *p* < 0.01 and + *p* < 0.05 vs. LPS-treated cells. (**C**) Expression of CD14 on cell surface was evaluated by flow cytometry analysis. The graph represents MFI fold change values of CD14 normalized to untreated control cells. *** *p* < 0.001 vs. controls; and + *p* < 0.05 vs. LPS-treated cells. (**D**) Expression of YM-1 was evaluated by Western blot. (**E**) Representative photomicrographs showing changes in cell morphology after LPS stimulation in the presence or absence of NaE. Cells were photographed using a Nikon Eclipse TS100 microscope (200× and 400× magnification). N = 3. Western blot bar graphs are in Appendix A.

**Figure 5 antioxidants-12-00232-f005:**
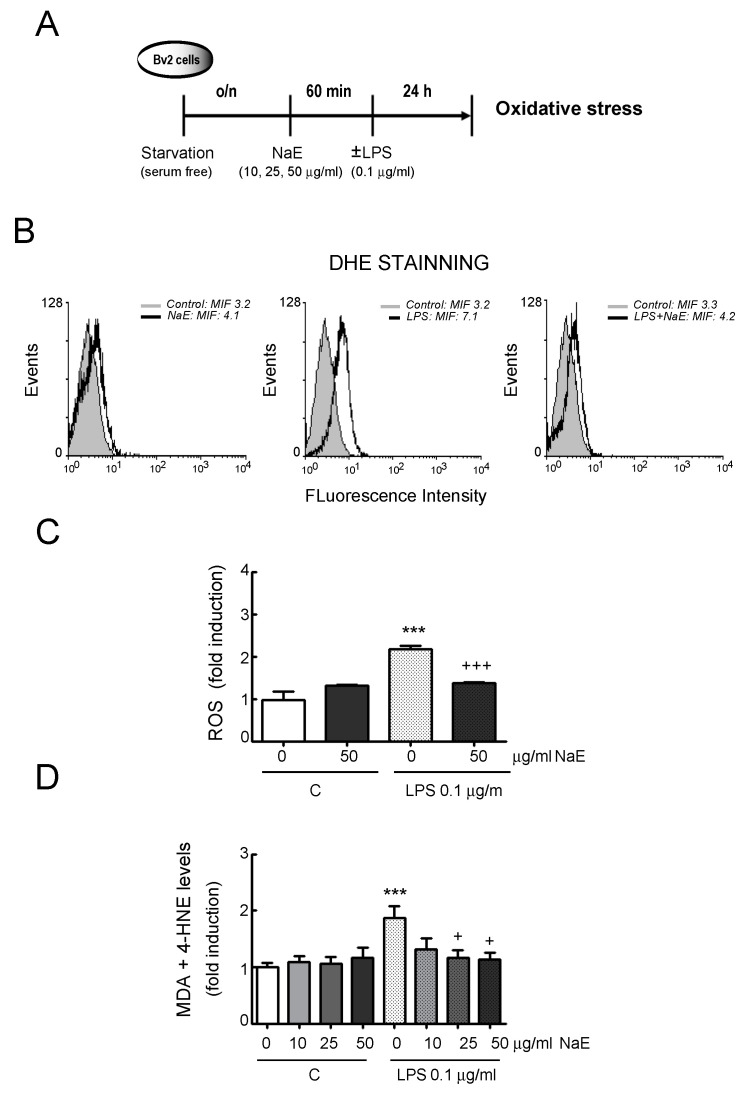
NaE extract modulates LPS-induced oxidative stress in BV2 microglia. (**A**) Experimental steps. Cells were pre-treated with the indicated doses of the NaE extract for 1 h and then stimulated or not with 0.1 μg/mL of LPS for 24 h. (**B**) After treatments, cells were incubated at 37 °C in the dark for 20 min with a culture medium containing 20 μM DHE to monitor intracellular superoxide anion production using flow cytometry. Representative histograms. (**C**) Quantified levels of ROS. (**D**) Lipid peroxidation (MDA + 4-HNE) levels. Values are expressed as the mean ± SD of the three independent experiments. *** *p* < 0.001 vs. controls; and +++ *p* < 0.001 and + *p* < 0.05 vs. LPS-treated cells.

**Figure 6 antioxidants-12-00232-f006:**
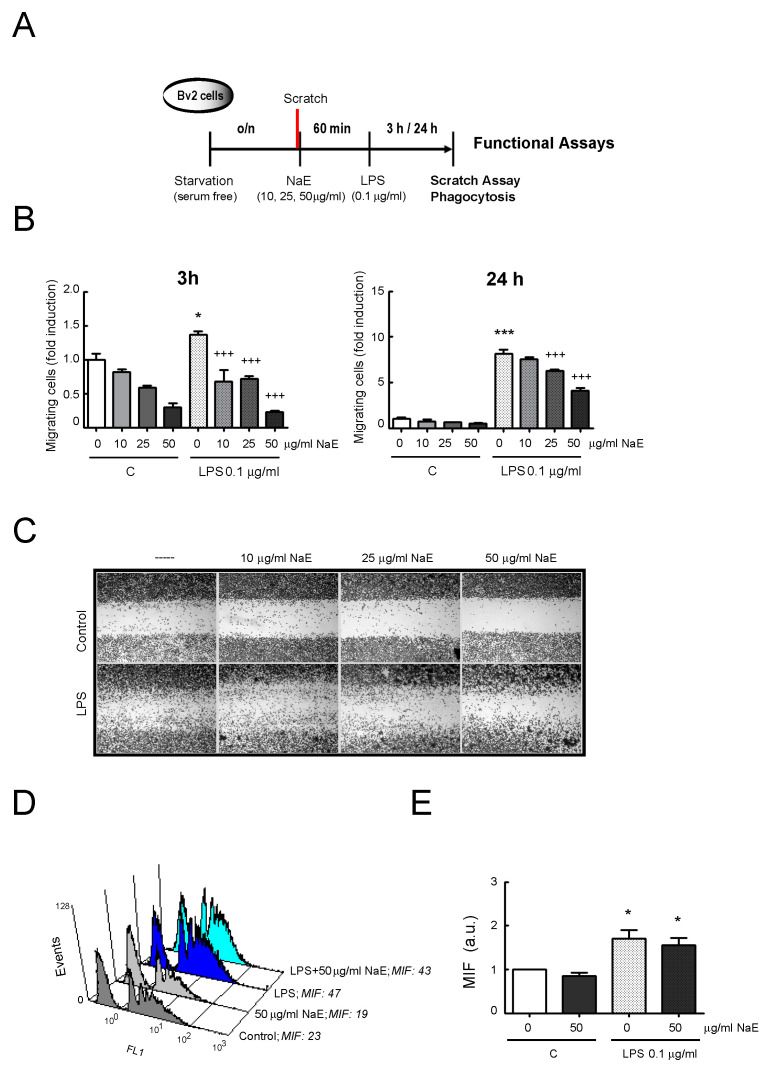
Effect of NaE extract on BV2 microglia functions. (**A**) Experimental steps. BV2 cells were pre-treated with the indicated doses of the NaE extract for 1 h at 37 °C and then stimulated or not with 0.1 μg/mL of LPS. (**B**) Migration assay: Data were reported as BV2 cell migration into the scratch area at 3 h and 24 h post-scratch. Values are means ± SD of the three independent experiments. *** *p* < 0.001 and * *p* < 0.05 vs. control; and +++ *p* < 0.001 vs. LPS-treated cells. (**C**) Representative images showing cell migration in the different conditions at 24 h. Magnification, 40×. (**D**,**E**) Phagocytosis assay using YG-microparticles and flow cytometry analysis. (**D**) Representative histogram. MFI: mean fluorescence intensity. (**E**) Graph represents fold change in mean fluorescence intensity (MFI) of YG-microparticles phagocytized calculated with respect to unstimulated controls cells. * *p* < 0.05 vs. control cells. N = 3.

**Figure 7 antioxidants-12-00232-f007:**
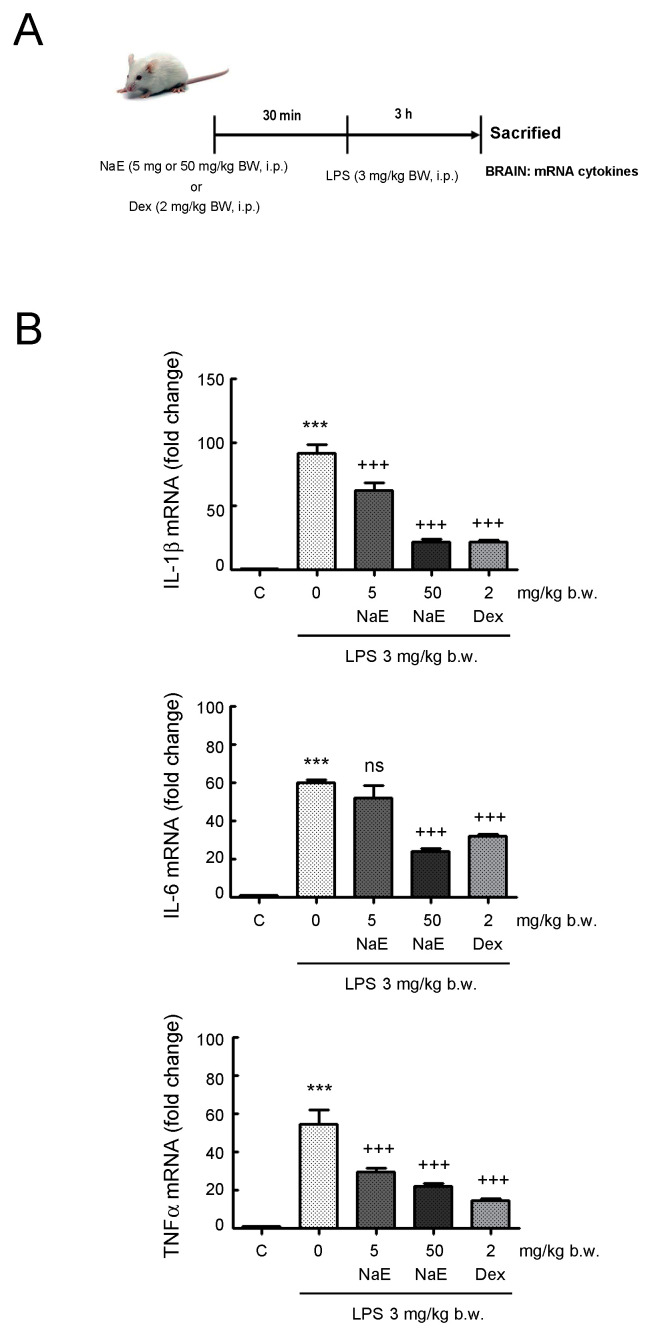
Effect of NaE extract on LPS-induced neuroinflammation in mice. (**A**) Experimental steps. Mice were treated with vehicle, dexamethasone (2 mg/kg) (Dex), NaE extract (50 mg/kg) or NaE extract (5 mg/kg) 30 min prior to intraperitoneal injection of LPS (3 mg/kg). (**B**) IL-1β, IL-6 and TNF-α mRNA was determined in brain tissue at 3 h after i.p. LPS injection by RT-qPCR analysis. Values were calculated using the ∆∆Ct method normalized to GAPDH and the control group (only vehicle administration) and expressed as mean ± SD (n = 6). *** *p* < 0.001 vs. the control group; and +++ *p* < 0.001 vs. the LPS-treated group.

## Data Availability

Data are contained within the article or Appendix A.

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
