# Peer review of "Anti-Neuroinflammatory Potential of a Nectandra angustifolia (Laurel Amarillo) Ethanolic Extract"

_antioxidants, 2023, doi:10.3390/antiox12020232_

Round 1
Reviewer 1 Report
The manuscript is well-designed and written, congratulation.
I can only highlight a couple of minor notes:
line 97: CO2 should be CO2
line 322: decrease should be decreased
I agree that in vitro and in vivo, being original Latin terms, should be written in italic as it was well done in the abstract.
Reviewer 2 Report
The present study aims at investigating in vitro and in a mice model the potential benefits of a Nactalia Angustifolia extract (NaE) against the pathogenic inflammatory activation of microglia cells.
The work is interesting since it points to NaE, a natural source of polyphenols used for its anti-oxidant and anti-inflammatory properties in South America folk medicine, as a novel, not yet deeply investigated phytochemical with bioactivity promising for the clinical application against neuroinflammatory events associated with neurodegeneration.
As clearly reported, the authors have demonstrated NaE capability to counteract microglia BV2 cells pro-inflammatory polarization by reducing ROS production and cytokines secretion induced by LPS in this experimental model, and improving cell migration without affecting phagocytosis, as well as to ameliorate in vivo neuroinflammation.
Nevertheless, several important aspects need a more in-depth investigation to explain and support NaE protective potential suggested by the authors. A point-by-point list of them is reported below:
1- the link between the redox-sensitive inflammatory pathways (NF-kB, MAPKs) and NaE activity is not clear. Moreover, why did the authors evaluate P70S6K, a mediator in the PI3K/mTOR signaling, a pathway involved, inter alia, in cell survival and autophagy? Undoubtedly this pathway is of main importance in neurodegeneration, but, without any other evaluation associated to it, what could the herein results suggest? Did the authors aimed at finding a possible interconnection between P70S6K and the other kinases? Experiments with inhibitors specific for the mediators target of NaE are needed to elucidate whether and how it affects pro-inflammatory and pro-oxidant microglia activation by their regulation;
2- NaE seems to counteract oxidative burst in microglia, but it seems also not effective towards SOD, HO-1, and Keap1 (see also “Discussion”, lines 479-486). Please also remember that Keap1/Nrf2 pathway is one of the MAPKs up-stream inducers. How do the authors explain this apparent contradiction? To confirm and elucidate NaE antioxidant properties and to identify the overall signal cascade possibly triggered by it some other factors and/or enzymatic systems responsible for redox state have to be investigated (e.g. p21- and p26-dependent Nrf2 expression, or NOX, ….);
3- why did the authors choose 24 h treatment for ROS evaluation? This end-point time is likely right to see lipoperoxidation products but I wonder if it is too long to see also highly reactive species such as superoxide. In this connection, H2O2 have not been detected, contrary to what is stated at line 440. Moreover, as regards MDA and HNE, where they have been quantified, in total homogenate? In the surnatant after centrifugation? It is important, since lipoperoxidation occurs in membranes;
4- in Fig 4.B, Il-6 and TNFα levels in control cells are very low, they even seem under detection limit! Was it possible to do statistical analysis?
5- in the most of “Results” it is not clear to which group comparison p values refer, it must be defined. For this reason, it is also recommendable to report for all the results the histogram graph to better depict their trend and statistical. In particular histograms are necessary for Fig. 6.D since the present graphical representation do not allow to deduce flow cytometry results;
6- the quality of figures 1.C and 4.E is bad thus it is not possible to understand the actual results. It could be useful to magnify them.
7- in “Discussion”, author statement at line 480 is improper since in this work NaE has been proven to affect at least some features of LPS-induced response in microglia cells.
In addition, there are some minor considerations:
8- in “Material and Methods”: for the sake of clarity, please add NaE chemical composition in 2.2; in 2.3, at lines 101-102, specify that treatment schemes are reported in figures; in 2.6, at line 125, describe blocking buffer composition; in 2.14 specify primer Company and add reference for the DDCt method;
9- in Fig. 6.C, I suggest to add also images of cell migration at 3 h;
10- ref. 13 is not available neither in PUBMED nor in other Internet sources, it should be replaced by a reference commonly attainable;
11- some abbreviations have not been introduced the first time they appeared in the text; please also note that sometimes the authors use RT and other “room temperature”
12- orthographic and in particular punctuation faults or lack (e.g. capital instead of uppercase letters, comma misuse);
13- few English mistakes, in particular word misuse (e.g. at line 102, several instead of different);
The authors have to keep in consideration these observations to proceed with the manuscript publication.
Reviewer 3 Report
In this manuscript, the Authors investigate the antiinflammtory potential of an ethanolic extract of the plant Nectandra angustifolia (NaE) in a microglia-like cell like and an a mouse model of neuroinflammation. In vitro, they demonstrated that pretreatment with NaE greatly reduces the expression of inflammation markers such as NLRP3, pERK, ROS and peroxidated lipid as well as the release of pro-inflammatory cytokines such as TNFa and IL6 induced by LPS challenge. Notably, NaE pretreatment sustains the release of the antiinflammatory cytokine IL10, which would be otherwise obliterated by LPS. Functionally, NaE pretreatment reduces the migration property of the cells both under basal and upon LPS stimulation. In vivo, pretreatment with NaE prior to injection with LPS reduces the induction of pro-inflammatory cytokines in the mouse brain as efficiently as the known-antiinflammatory drug Dexamethason. Overall the paper is very interesting and well written. Few minor suggestions:
1) please specify that it is an ethanol extraction already in the abstract
2) Fig1C, please provide zoomed in pictures to appreciate the morphology (as in Fig4C)
3) western blot should be quantified
4) please consider to include a scheme at the end to visually wrap up their in vitro mechanistic findings
5) are the Authors planning to test whether NaE administration could be effective also post-exposure to LPS or in models of chronic neuroinflammation? this would further validate the therapeutic potential of NaE and the info should be perhaps included in the discussion
Author Response
Please see the attachement

Reviewer 4 Report
The manuscript by Crescitelli et al. investigates the anti-neuroinflammatory potential of Nectandra angustifolia ethanolic extract (NaE) in both in vitro and in vivo models. A previous study from the same authors was focused on characterizing NaE chemical content and highlighting its antioxidant and anti-inflammatory properties. In the present study, the authors show that NaE pretreatment of LPS-activated BV2 microglial cell line determined a modulation of proinflammatory mediators, as well as reduced levels of iNOS and COX-2 proteins. In addition, NaE pretreatment determined an overall reduction in oxidative stress levels, while promoting the expression of M2 state markers. Lastly, the authors claimed that NaE is able to modulate the neuroinflammation in vivo through a marked reduction in proinflammatory cytokines mRNA levels.
Overall, the manuscript reports some interesting evidence related with an anti-neuroinflammatory effect of NaE on a preclinical level. However, there are some important concerns that need to be assessed before the manuscript acceptance.
The manuscript shows some writing mistakes. Please check grammar, spelling, punctuation, typographical emphasis, abbreviations through all the paper, in order to be consistent through all the paper.
Major comments:
1) In the section 3.1 of Results, from line 241 to line 247 it’s reported that NaE pretreatment is able to modulate the proinflammatory signaling pathway, determining a significant inhibition (p < 0.05) in the phosphorylation of Erk1/2, JNK1/2 and p38 MAPKs, as well as P70S6K. However, there’s no graph reported the WB quantification in the relative figure (Figure 2). Please include all the graphs with the relative statistical analysis.
2) The authors claimed that NaE pretreatment of mice could modulate neuroinflammation caused by intraperitoneal LPS injection thanks to a significant reduction in proinflammatory cytokines mRNA levels. However, the sample size per group is not enough wide to clearly state that NaE pretreatment could ameliorate neuroinflammation in an in vivo model, even if there’s a strong statistical significance. The authors should increase the sample size to 5 mice per group at least. Moreover, a significant reduction in proinflammatory cytokines mRNA levels is not enough to state the anti-neuroinflammatory effect of NaE in vivo. The authors should perform several additional experiments to prove this aspect, i.e. ELISAs for pro-inflammatory cytokine, characterization of microglial activation state (M1 and M2 markers profile and microglia morphology).
3) The doses of NaE extract are not consistently used among the diverse experiments in BV2 cells. Please justify this point.
4) The use of dexamethasone (as positive control?) should be described and motivated.
5) The overall quality of the figures is low, making difficult for the reader to understand graphs and representative photomicrographs. Please improve this aspect (especially for Figure 2C, Figure 3C, Figure 4B-E, Figure 5B-D, Figure 6, Figure 7). Also, captions do not clearly describe the related figures and more informations need to be included for both western blot images and photomicrographs (for example: Representative photomicrographs showing differences in BV2 microglial cell morphology when activated by LPS 0.1 µg/ml and pretreated with NaE at different concentrations (25, 50 µg/ml). Scale bar: x µm).
6) In materials and methods section, the author need to implement the given information in order to make the experiments reproducible by other scientist. For example, the protocols for protein extraction and ELISA need to be described in details and specifics about the instruments used, or the way to present the data.
7) The introduction and discussion section appear not enough detailed and need to be improved. For example, the role of NSAID in clinical trials and, in general, of anti-neuroinflammatory drugs, the M1 vs M2 microglial phenotype, as well as the data regarding Nectandra angustifolia ethanolic extract effects need to be more described and discussed.
8) The references appear to be not sufficient, in particular in the introduction and discussion (i.e. line 54, 425, 490, 492)
Minor comments:
1) The title of the manuscript should be corrected from “Anti-Neuroinflammatory potential of a Nectandra angustifolia (Laurel Amarillo) ethanolic extract in in vivo and in vitro” to “Anti-Neuroinflammatory potential of a Nectandra angustifolia (Laurel Amarillo) ethanolic extract in vivo and in vitro”.
2) In Material and Methods section, authors should state how LPS and dexamethasone solutions were prepared (Which was the vehicle? Saline or DMSO or ethanol?). Moreover, it is important to add information regarding the calculation of the doses used in the animals.
3) Please be consistent with specific terms throughout the manuscript (i.e. pre-treatment or pretreatment; pro-inflammatory or proinflammatory; TNF-α or TNFα). Moreover, please check the abbreviations, that sometimes are missing or not spell out when used for the first time (i.e. Ho-1, iNOS etc).
4) Line 308, 328, 470, 515: no treatment but pretreatment. Line 407: dexamethasone pretreatment, not treatment.
5) In Figure 7 caption, line 414: DDCt should be replaced with ΔΔCt, as reported in the section 2.14 of Material and Methods.
6) Line 72 and 419, “to the best of our knowledge” need to be eliminated or rephrase.
7) The starvation period for BV2 experiments is 24h, as indicated in methods, or o/n, as described in the figures? Please modify them in accordance with the correct value.
8) Figure 6D, the graph reporting the flow cytometry data should be replace with another type of more explicative one.
9) Figure 7A, “3days” should be modified in “3h”
Reviewer 5 Report
This study evaluated the therapeutic effects in vitro (BV2 microglial cultures) and in vivo (mice) of Nectandra augustifolia (NaE) on the progression of neuroinflammation signaling mediators. The authors report that NaE had anti-neuroinflammatory properties by reducing the induction of proinflammatory mediators, attenuating lipopolysaccharide (LPS)-induced reactive oxygen species (ROS) and lipid peroxidase build-up, and affecting other mechanism markers.
The abstract appropriately summarized the findings of this study. The introduction appropriately described the rationale of the study, although appropriate citations are missing as indicated below. The methods are describe appropriately, although appropriate citations should be added as indicated below. The results section needs some work including the need to add bar graphs of the quantification of the Western blots to Figure 2 (B, C), Figure 3 (B, D, E), and Figure 4D, and provide F and p values where appropriate throughout this section. The lack of F and p values in many places is a big concern. The discussion is appropriate although many citations are missing as detailed below. The Conclusion is appropriate.
Detailed comments:
General:
Throughout the manuscript BV2 and BV-2 are used inconsistently.
Abstract:
Line 22: Remove comma after analgesic.
Line 26: Add a comma after cells.
Line 31: remove comma after as well as.
Line 34: Specify whether migration was increased or decreased.
Line 36: Add “in” between beneficial preventing.
Introduction:
Line 44: Add appropriate citations at the end of the sentence.
Line 46: Remove “The” from the start of the sentence.
Line 48: Add appropriate citations at the end of the sentence (…tissue.).
Line 53: Change aggravates to aggravate.
Line 55: change NSAIDs to non-steroidal anti-inflammatory drugs (NSAIDs).
Line 62: Add appropriate citations at the end of the sentence.
Line 64: With Northeastern, do the authors mean the eastern parts of Argentina, Brazil, and Uruguay?
Line 63: add (NaE) after Mart.
Lines 73 and 75 replace Nectandra augustifolia (NaE) with NaE.
Materials and Methods:
Line 86: Remove the space in front of the sentence.
Line 90: Replace “submitted to” with under.
Line 92: Replace “have been already” with have already been.
Line 92: Add “to” between prior LPS.
Line 104: Remove comma after kit.
Line 114: This paragraph would benefit from appropriate citations that used similar methods for the flow cytometric analysis.
Line 116: Add ”a” between with PE-.
Line 117: Change “500xg” with 500 x g.
Line 118: Add “in” between suspended PBS-.
Line 124: Add “a” between After 4.
Line 125: Replace RT with room temperature.
Line 125: Remove extra spaces before they.
Line 121: This paragraph would benefit from appropriate citations for methods used.
Line 135: Briefly explain what “the enhanced chemiluminescence procedure” is.
Line 137: This paragraph would benefit from appropriate citations for methods used.
Line 144: This paragraph would benefit from appropriate citations for methods used.
Line 153: This paragraph would benefit from appropriate citations for methods used.
Line 159: This paragraph would benefit from appropriate citations for methods used.
Line 176: This paragraph would benefit from appropriate citations for methods used.
Line 189: Please clarify what “during adaption period” means.
Line 194-195: Please switch the 50 mg and 5 mg/kg NaE groups in the sentence so the order represents the order in the relevant figures.
Line 201: This paragraph would benefit from appropriate citations for methods used.
Line 216: Replace “ANOVA” with analysis of variance (ANOVA).
Results:
Line 226: Please provide the F and p values at the end of the sentence.
Line 230: Please provide the F and p values at the end of the sentence.
Line 230: Change “with NaE effect” to with the effect of NaE.
Line 244-245: Replace “Figure 2B, showed” with Figure 2B shows.
Line 247: Please provide the F value.
Line 253: Please provide the F and p values where appropriate.
Line 260-261: Delete the sentence “Total…M&M” as this is a duplication.
Line 258: The actual quantification of the Western blot results should be shown with additional bar graphs in Figure 2 for each of the measured proteins. The authors must have used this data to statistically evaluate the results. Quantification could be optical density of the bands corrected for background staining.
Line 267: NO is described as nitric oxide here, but this should be done the first time it occurs in the manuscript.
Line 295: The actual quantification of the Western blot results should be shown with additional bar graphs in Figure 2 (B, D) for each of the measured proteins. The authors must have used this data to statistically evaluate the results.
Line 271: Include appropriate F and p values at the end of the sentence (…(Figure 3B).).
Line 272: Include appropriate F and p values at the end of the sentence (…factors.).
Line 275: Add F value.
Line 276: Include appropriate F and p values at the end of the sentence (…(Figure 3C).).
Lines 280-284: Please include appropriate F and p values where appropriate as indicated in detail above.
Line 287: Please include appropriate citations at the end of the sentence (…tissues.).
Lines 285-291: Please include appropriate F and p values where appropriate as indicated in detail above.
Line 306: Please include appropriate F values for each protein tested.
Line 308: Please include appropriate F and p values for IL-10.
Line 322: Change “decrease” to decreased.
Line 327: Please include appropriate F and p values.
Line 330: Please include appropriate F and p values. Also, add (Figure 4D) to the sentence.
Line 332: Please add appropriate citations after signals.
Lines 336-339: Was this assessed qualitatively or quantitatively? Personal observation not really assessed?
Lines 343-345: Please include appropriate F and p values for each measure.
Line 344: How was the flow cytometry quantified?
Line 347: Please include appropriate F and p values.
Line 361: Please add appropriate citations after neurodegeneration.
Line 367: Please include appropriate F value.
Line 370: Please include appropriate F and p values after …microglial cells.
Line 388: delete the second “we”.
Line 391: How was phagocytotic capacity quantified?
Line 392: Please include appropriate F value.
Line 398: Please add appropriate citations after of microglia.
Line 404: Please include appropriate F and p values.
Line 405: Please include appropriate F and p values.
Line 407: Please include appropriate F and p values.
Discussion:
Line 424: Add “treatments” between analgesic and.
Line 424: Add appropriate citations at the end of the sentence (…others.).
Line 425: replace “specie” with species.
Line 427: Please put a comma before and after “including flavonoids and phenolic acids.”
Line 428: Remove the period after NaE.
Line 429: Add appropriate citations after …disorders.
Line 433: Add appropriate citations after …CNS.
Line 437: Add appropriate citations after …effects.
Line 479: Add a comma after In this study.
Line 484: Replace “into” with in.
Lines 485-486: Please delete “,gaining…future” as it does not add to the discussion.
Lines 487-488: Add appropriate citations.
Line 490: Add appropriate citations at the end of the sentence.
Lines 509-517: Please write this section in the past tense.
Line 513: Please remove the comma after NaE.
Round 2
Reviewer 2 Report
Please, see the attached file

Reviewer 4 Report
The second version of the manuscript by Crescitelli et al. has been improved and the authors addressed most of the suggestions of the reviewer.
Nonetheless, some minor points should be considered by the authors.
1) Regarding the numbers of mice used for in vivo experiments, since the data are representative of 2 trials performed under the same conditions with similar results, the figure must include the results from n=6 mice ( and not just n=3). Please recalculate the values and modify the manuscript accordingly (figure 7 and caption).
2) The authors has included the graphs of WB quantification in the supplementary material to avoid overload of the manuscript. However, WB results without the graphs appear to be more difficult to understand. In ordere to increase the overall quality of the manuscript, please add the graphs reported in supplementary S2-S3-S4 to the corresponding figures.
3) In the Supplementary Figure 3 (S3), the unit of measure of NaE extract (μg/ml NaE) hasn’t been reported on the SOD2/β-actin graph. Please, correct it.
4) Please, check the position and typographical emphasis of each section titles (i.e. line 154, 172).
5) Please, check all the references in the “References” section and report them in a consistent way. (i.e. lines 656,657; lines 675,676; lines 683-685 are reported differently from other references).
6) In line 153, what does “(Referencia)” stand for?
Reviewer 5 Report
Overall the manuscript has significantly improved. Just a few editorial suggestions:
Line 72: please change “revision” to review.
Line 225: please add “to the” between prior LPS.
Line 339: please replace “on” to in.
Line 433: please replace “causes” with caused.
Line 507: please change “is to note” to is of note.
Lines 529-530: Please change to:
How-ever, the role of NaE modulating the autophagy flux needs supplementary exploration. Interestingly, positive effects of a variety of plant extracts…
Line 551: please replace “at” with in the.
Line 571: please change “being F-actin” to F-actin being.
